# Predicting the potential global distribution of *Ixodes pacificus* under climate change

**Fengfeng Li** [1], **Qunzheng Mu**[2☯], **Delong Ma**[2☯], **Qunhong Wu**[3]*

**1** School of Public Health, Shandong Second Medical University, Weifang, Shandong, People's Republic of China, **2** State Key Laboratory of Infectious Diseases Prevention and Control, National Institute for Communicable Disease Control and Prevention, Chinese Center for Disease Control and Prevention, Beijing, People's Republic of China, **3** Department of Social Medicine, Health Management College, Harbin Medical University, Harbin, China

☯ These authors contributed equally to this work.
* wuqunhong@163.com

**Data Availability Statement:** All relevant data are within the manuscript and its Supporting Information files.

**Funding:** The author(s) received no specific funding for this work.

## Abstract

In order to predict the global potential distribution range of *Ixodes pacificus* (*I. pacificus*) under different climate scenario models in the future, analyze the major climate factors affecting its distribution, and provide references for the transformation of passive vector surveillance into active vector surveillance, the maximum entropy model (MaxEnt) was used in this study to estimate the global potential distribution range of *I. pacificus* under historical climate scenarios and different future climate scenarios. The global distribution data of *I. pacificus* were screened by ENMtools and ArcGIS 10.8 software, and a total of 563 distribution data of *I. pacificus* were obtained. Maxent 3.4.1 and R 4.0.3 were used to screen climate variables according to the contribution rate of environmental variables, knife cutting method and correlation analysis of variables. R 4.0.3 was used to calculate model regulation frequency doubling and feature combination to adjust MaxEnt parameters. The model results showed that the training omission rate was in good agreement with the theoretical omission rate, and the area under ROC curve (AUC) value of the model was 0.978. Among the included environmental variables, the Tmin2 (minimum temperature in February) and Prec1 (precipitation in January) contributed the most to the model, providing more effective information for the distribution of *I. pacificus*. MaxEnt model revealed that the distribution range of *I. pacificus* was dynamically changing. The main potential suitable areas are distributed in North America, South America, Europe, Oceania and Asia. Under the future climate scenario model, the potential suitable areas show a downward trend, but the countries and regions ieeeeeeenvolved in the suitable areas do not change much. Therefore, the invasion risk of the potential suitable area of *I. pacificus* should be paid attention to.

## Introduction

*Ixodes pacificus* (*I. pacificus*), also known as the western blacklegged tick, is occurs mainly in western North America. It can carry Anaplasma phagocytophilum, Borrelia burgdorferi sensu stricto and Borrelia miyamotoi and other zoonotic pathogens are major vectors of Lyme disease

**Competing interests:** The authors have declared that no competing interests exist.

in the western United States [1, 2]. Since 1950, the historical geographic range of *I. pacificus* has been expanding due to human activities, climate, and land utilization. However, its distribution range has been relatively stable in the past 20 years, and it is possible that the region where *I. pacificus* is now found has almost reached the level of its basic ecological niche [3].

Since the last century, with the rapid increase of the global population and the increasing frequency of human economic activities, the global climate has undergone significant changes [4]. According to the Sixth Assessment Report of the United Nations Intergovernmental Panel on Climate Change (IPCCAR6), global land and ocean temperatures have increased by 1.09° C between 1901 and 2020. Global climate change has a serious impact on the global natural ecosystem, changing the geographical and natural environment, destroying biodiversity, and even threatening human life and health [5, 6]. Among them, the distribution and quantity of vectors are easily affected by temperature, and the propagation and spread of pathogens in their bodies are also affected by climate factors [7]. Global warming and changes in precipitation provide environmental advantages for the invasion and colonization of species. It is possible for *I. pacificus* to spill over and colonize, which will pose a huge threat to local human and animal health and public health.

Ecological niche modeling (ENM) is a mathematical model that estimates the actual and potential distribution of target species by quantifying the relationship between the distribution data of species and the corresponding environmental variables [8]. ENM can be divided into correlative models (MaxEnt, BIOCLIM, HABITAT and DOMAIN) and mechanistic models (BOMAIN and CLIMEX) according to basic principles [9, 10]. MaxEnt is a species distribution prediction model based on the theory of maximum entropy. It can calculate the contribution rate of environmental variables using the knife cutting method and evaluate its own model using the area under ROC curve (AUC). It is a model with more extensive application and better prediction ability among species distribution models [11, 12].

Based on the historical distribution data and climate data of *I. pacificus*, the maximum entropy model (MaxEnt) was used to estimate its global distribution range under historical climate scenarios and different future climate scenarios, and the influence of environmental climate factors on the change of distribution range was analyzed.

## Materials and methods

### Distribution data collection and processing of *I. pacificus*

The occurrence points of *I. pacificus* were acquired mainly from the Global Biodiversity Information Facility (GBIF; GBIF.org (29 March 2024) GBIF Occurrence Download https://doi.org/10.15468/dl.3wrhb5) database and related research literature of CNKI, Web of Science, PubMed, MEDLINE, Embase, and other databases. The time range is from 1950 to 2024. Screen the field survey records of *I. pacificus* and select the latitude and longitude information of the collection points (S1 File). Via OpenStreeMap (https://osm.openmaptiles.org/, accessed on 31 March 2024), the accuracy of coordinate points were checked and points with apparent errors in geographical coordinates were removed. To avoid overfitting, use ENMtools software to remove redundant data within the same grid. All sample point data were saved in a CSV file, which contained three columns, i.e., species name, longtitude and latitude coordinate values.

### Software and geographic data

Steven J. Phillips, Miroslav Dudík, Robert E. Schapire. [Internet] Maxent software for modeling species niches and distributions (Version 3.4.1). Available from url: http://biodiversityinformatics.amnh.org/open_source/maxent/. Accessed on 1 March 2024. ArcGIS

software (version 10.8) was purchased by the Vector Control Department of the Institute of Infectious Disease Control and Prevention, Chinese Center for Disease Control and Prevention. R software (version 4.0.3, http://www.r-project.org/, accessed on 31 May 2021) and DIVA-GIS software (version 7.5.0, https://www.diva-gis. org/, accessed on 31 May 2021) were used to adjust model parameters. ActivePrel (version 5.26) was downloaded from https://www.activestate.com (accessed on 12 May 2023). The base map of the world was derived from the 1:110 Natural Earth I (http://www.naturalearthdata.com, accessed on 11 October 2021) drawn by the North American Cartographic Information Society (NACIS). The scale of the map in this article represents mid-latitude distances.

## Environmental data

Download from WorldClim website (https://worldclim.org/) Including historical climate and environmental monitoring data (1970–2000) and future climate and environmental prediction data, a total of 56 climate variables (at a 2.5 min spatial resolution): 19 bioclimatic variables (bio 1–19) (Table 1), 36 climate variables (monthly maximum temperature, monthly minimum temperature, and monthly precipitation), and 1 elevation data. The future climate data includes climate variables from 2021 to 2100 using the BCC-CSM2-MR medium resolution climate system model of the China (Beijing) Climate Center climate system in the 6th International Coupled Model Comparison Program (CMIP6), which includes four shared social economic paths (SSP) from 1–2.6 (sustainability and radiative forcing of 2.6 $W/m^2$ to 2100), 2–4.5 (middle of the road and forcing of 4.5 $W/m^2$ to 2100), 3–7.0 (regional rivalry and forcing of 7.0 $W/m^2$ to 2100), and 5–8.5 (fossil-fuel development and forcing of 8.5 $W/m^2$ to 2100). Use the ArcGIS 10.8 software conversion tool to convert it to ASCII format for software use.

To avoid overfitting of environmental data, it is necessary to filter the environmental data. Related studies have shown that environmental variables such as bio8 (average temperature of the wettest season), bio9 (average temperature of the driest season), bio18 (precipitation of the

**Table 1. Descriptions of each of the 19 bioclimatic variables.**

| Variable | Description | Unit |
| :---: | :---: | :---: |
| Bio1 | Annual Mean Temperature | ˚C |
| Bio2 | Mean Diurnal Range (Mean of monthly (max temp—min temp)) | ˚C |
| Bio3 | Isothermality (BIO2/BIO7) (×100) | % |
| Bio4 | Temperature Seasonality (standard deviation ×100) | ˚C |
| Bio5 | Max Temperature of Warmest Month | ˚C |
| Bio6 | Min Temperature of Coldest Month | ˚C |
| Bio7 | Temperature Annual Range (BIO5-BIO6) | ˚C |
| Bio8 | Mean Temperature of Wettest Quarter | ˚C |
| Bio9 | Mean Temperature of Driest Quarter | ˚C |
| Bio10 | Mean Temperature of Warmest Quarter | ˚C |
| Bio11 | Mean Temperature of Coldest Quarter | ˚C |
| Bio12 | Annual Precipitation | mm |
| Bio13 | Precipitation of Wettest Month | mm |
| Bio14 | Precipitation of Driest Month | mm |
| Bio15 | Precipitation Seasonality (Coefficient of Variation) | % |
| Bio16 | Precipitation of Wettest Quarter | mm |
| Bio17 | Precipitation of Driest Quarter | mm |
| Bio18 | Precipitation of Warmest Quarter | mm |
| Bio19 | Precipitation of Coldest Quarter | mm |

warmest season), and bio19 (precipitation of the driest season) may cause spatial illusions during modeling [13]. Therefore, 52 environmental variables are excluded and remain. Import 52 environmental variables into Maxent 3.4.1 software and load distribution point data. Use Jackknife method to analyze the importance of environmental factors and obtain their contribution. Remove environmental variables with a contribution rate of less than 1.0% to the model, and then use R software to perform Pearson correlation analysis on environmental variables with a contribution rate of ≥ 1%. Variables with an absolute correlation coefficient of less than 0.8 are retained and incorporated into the model. Variables with an absolute correlation coefficient of ≥ 0.8 are considered highly correlated, and the variable with the highest contribution rate is included in the model.

## Selection of model parameters

When there are few species distribution points, the model using default parameters may not be the optimal model [14]. To avoid overfitting of the model, we need to adjust the parameters of the model. all the collected distribution points of the *I. pacificus* were imported into the MaxEnt software, set to random seed 75% distribution point modeling and 25% distribution point verification modeling, the output format was "Cloglog", the output file type was "asc", the maximum iteration mode was to set select "Bootstrap" and the maximum number of repetitions was 5000. The number of repeated training rounds was set to 20 to reduce the uncertainty caused by outliers by applying threshold rule select "Minimum training presence".

We selected the best model by setting the feature combination (FC) and regularization multiplier (RM). The RM parameter was set to 8 levels: 0.5, 1, 1.5, 2, 2.5, 3, 3.5 and 4, and there were 5 feature combinations: automatic linear (Linear, L), quadratic (Q), fragmentation (hinge, H), product (P) and threshold (T). We manually set 5 characteristic parameters to obtain 8 features (L, LQ, LQP, QHP, LQH, LQHP, QHPT and LQHPT). To get the best model and control overparameterization, the R package "ENMeval" was used. The ENMeval package was implemented in R 4.0.3 "Checkerboard2" to search for the lowest delta value for Akaike's information criteria corrected for small sample sizes (AICc) to run the MaxEnt software on the candidate models.

## Evaluation of the model results

The area under the curve (AUC) value of the receiver operating characteristic curve (ROC curve) is a commonly used test method to evaluate the accuracy of species distribution models. The closer the training omission rate is to the theoretical omission rate, the higher the accuracy of the constructed model [15]. The evaluation standard of a ROC curve is: AUC value = 0.5–0.6, failure; 0.6–0.7, poor; 0.7–0.8, fair; 0.8–0.9, good; 0.9–1.0, excellent.

## Project the potential suitable area

Based on MaxEnt, the cloglog result output is an asc file. Using ArcGIS 10.8, the asc file is converted into raster data using the ASC Ⅱ to Raster conversion tool, and the Jenks natural breaks classification method is selected to reclassify the suitable habitat. The criteria are: 0–0.0539, unsuitable area; 0.0539–0.1941, low suitable area; 0.1941–0.4709, middle suitable area; 0.4709–1, high suitable area.

## Results

Global distribution points and environmental data of *I. pacificus*. After processing by ENMtools software, we obtained 567 points of *I. pacificus.* Using ArcGIS 10.8 to sample the

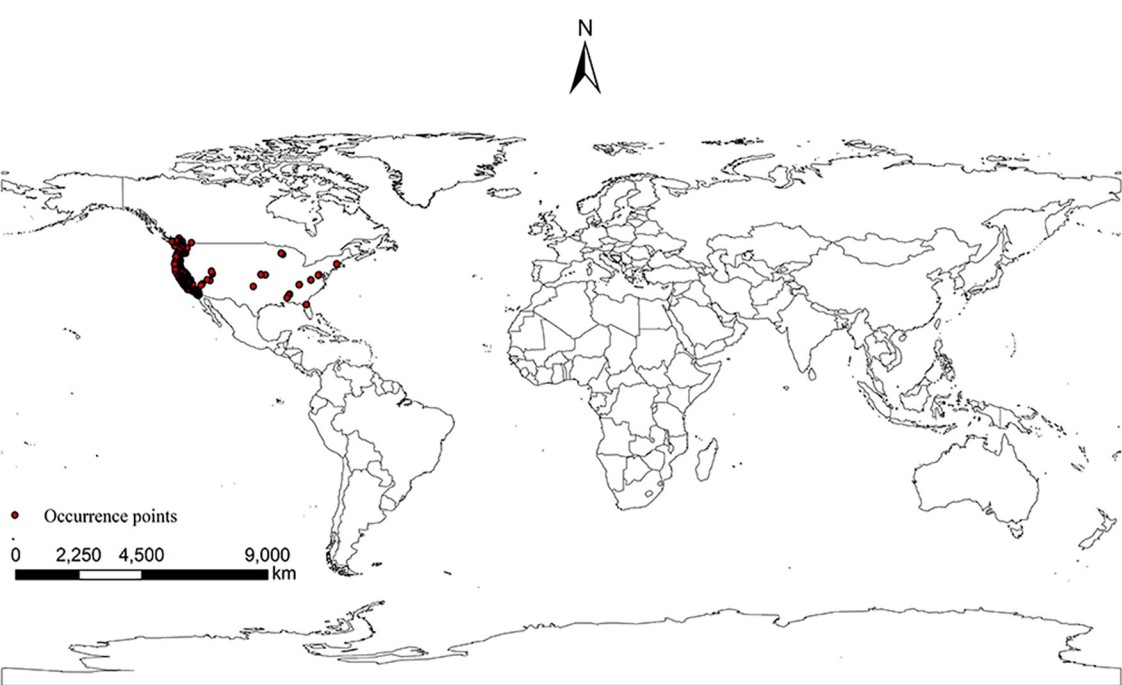

**Fig 1. Current distribution of *I. pacificus* in the world.** Made with Natural Earth. Free vector and raster map data @ naturalearthdata.com.

environmental data of distribution points, there were 4 coordinate positions where environmental data was not collected and were deleted, leaving 563 coordinate information (Fig 1). After analyzing the bioclimate and environmental variables using the Jackknife method, climate factors with a contribution rate of $\geq 1.0\%$ were selected, and Pearson correlation analysis was performed on the selected climate factors using R 4.0.3 (Fig 2). Finally, five climate factors were obtained for the model (Table 2).

## MaxEnt model parameter settings and accuracy evaluation

Optimize the model parameter settings using the ENMeval data package of R 4.0.3 software. The future combinations and regularization multiplier of the maximum entropy model in this study were LQHPT and 0.5 (Fig 3). In the MaxEnt model, the average omission rate of the test set is basically consistent with that of the training set (Fig 4). Under the current climate scenario conditions, the AUC value of the MaxEnt model is 0.978 (Fig 5), and the evaluation effect is "excellent". The model has high accuracy.

## The relationship between the distribution of *I. pacificus* and the environmental variables

We used the jackknife method to analyze the most important five variables that impact the distribution of *I. pacificus*. The results showed that Tmin2 (Minimum temperature of February) provided more effective information for the distribution of *I. pacificus*, and provided more information that was not included in other environmental variables, followed by Prec1 (Precipitation in January) (Fig 6). The response plot line between environmental variables and the probability of *I. pacificus* shows the average of 20 repeated runs of the model, and the blue margin shows the ± SD calculated after 20 repeated runs (Fig 7). The results show that the suitable

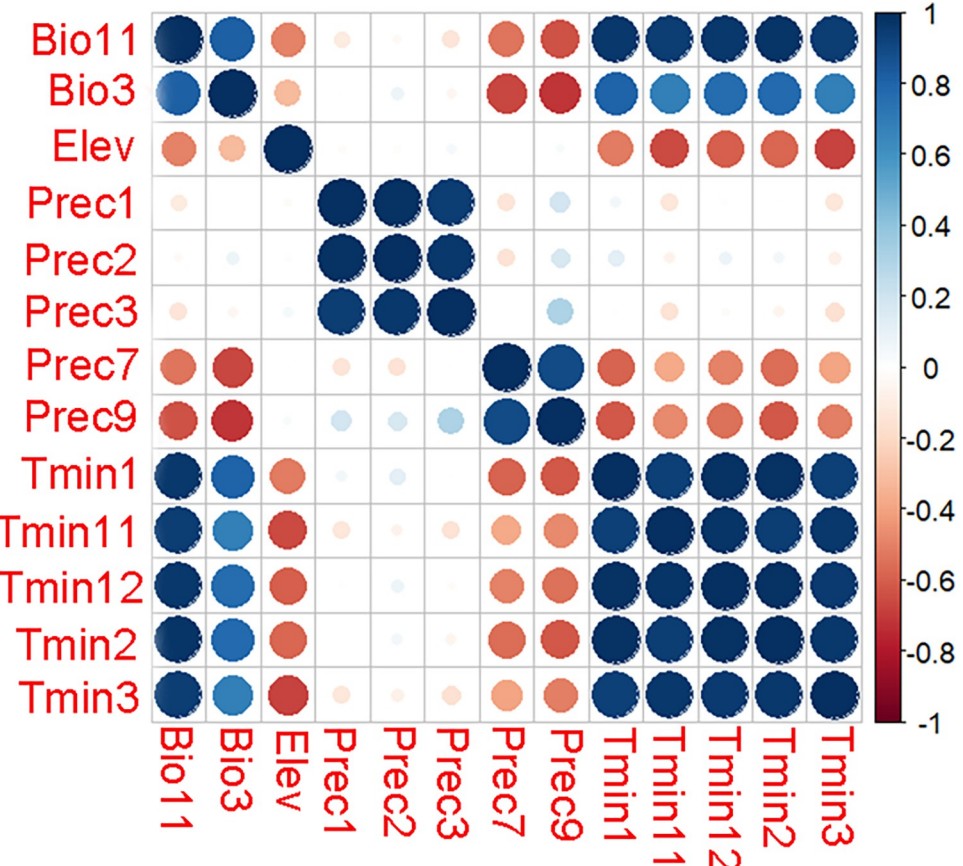

**Fig 2. Pearson correlation analysis of environmental variables.** The blue dots in the figure represent positive correlation, the red dots represent negative correlation, and the darker the color, the stronger the correlation.

range for Prec1 is between 70-626mm, with the most suitable precipitation being 300mm. When the precipitation is between 70-300mm, the probability of existence gradually increases. When the precipitation is between 327-626mm, the distribution probability has small fluctuations but is generally stable (Fig 7A). The suitable temperature for the Tmin2 (with a probability of $\geq 0.6$) is between -45.1–7.9°C. When the temperature is between -45.1–15.9°C, the distribution probability increases with the increase of temperature. When the temperature is between -15.9–1.7°C, the distribution probability fluctuates with an increase and a decrease, and there are three peaks in distribution probability. When the temperature is about -11.2°C,

**Table 2. Environment variables in projecting the distribution of *I. pacificus*.**

| Variables | Description | Unit | Contribution (%) |
|---|---|---|---|
| Prec1 | Precipitation in January | mm | 41.5 |
| Tmin2 | Minimum temperature of February | °C | 27.1 |
| Bio3 | Isothermality (BIO2/BIO7 × 100) (%) | mm | 24.1 |
| Prec7 | Precipitation in July | mm | 5.8 |
| Elev | Elevation | m | 1.4 |

Bio2, mean diurnal range (mean of monthly [max temp—min temp]); Bio7, temperature annual range (Bio5–Bio6); Bio5, max temperature of the warmest month; Bio6, min temperature of the coldest month.

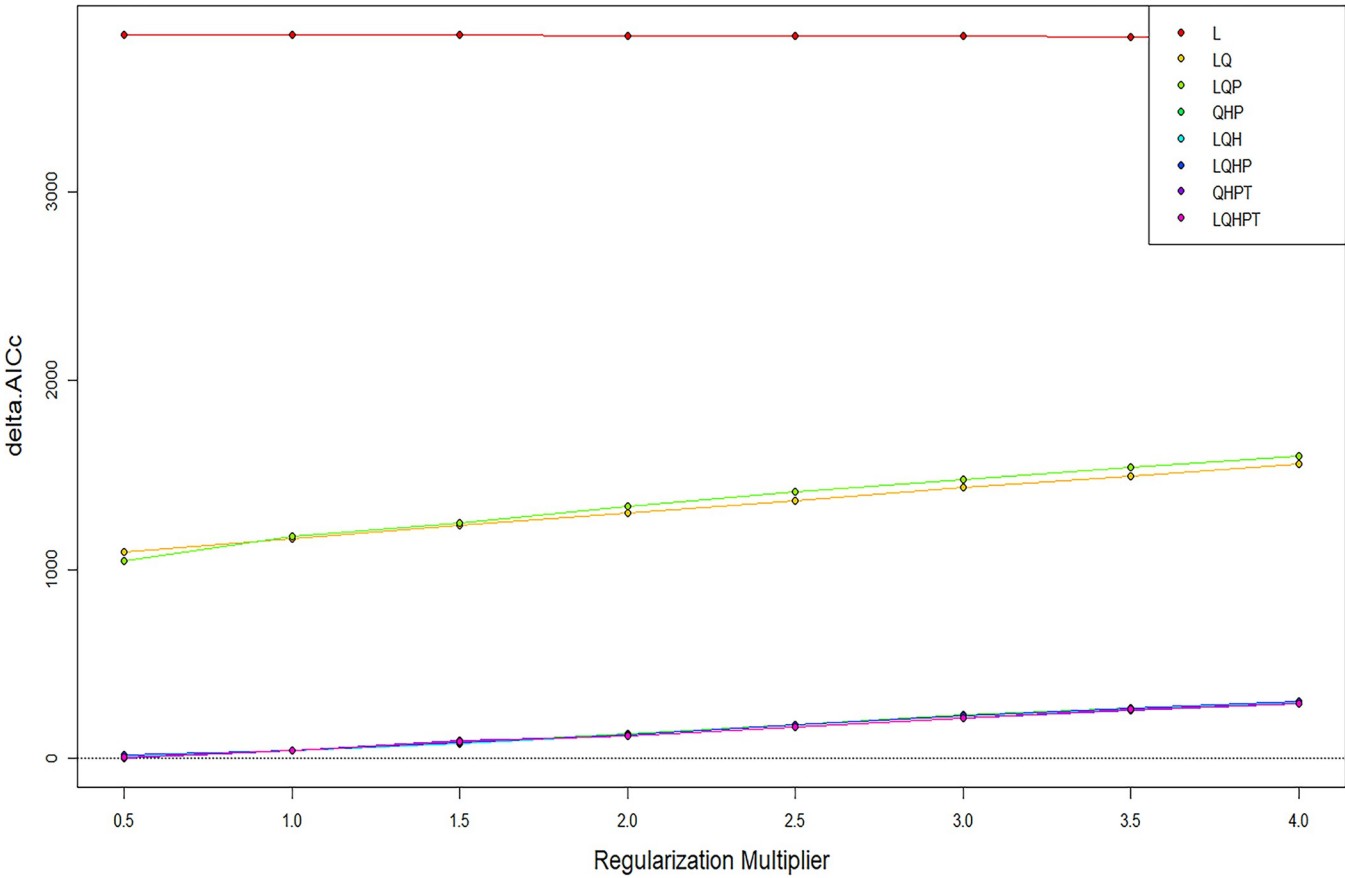

**Fig 3. AICc values for different regularization multiplier and feature combination.** Delta.

-8.7˚C, and -5.5˚C, respectively, the distribution probability decreases with the increase of temperature when the lowest temperature is between -5.5–7.9˚C (Fig 7B).

## The potential distribution of *I. pacificus* under near current climatic conditions

Under the near current climate and environmental conditions, based on the output results of the MaxEnt model, the near current suitable areas of the *I. pacificus* was reclassified using the ArcGIS 10.8 Jenks method. The suitable areas for *I. pacificus* is mainly concentrated in the western and eastern regions of the United States in North America, parts of southwestern Canada, parts of southern Greenland, and most of Iceland. The southern region of Chile in South America, western and southern parts of Argentina. Most countries along the Atlantic and Mediterranean coasts in western Europe, as well as some regions in central Romania, Hungary, Austria, Czech Republic, Poland, and other countries. Most of Lebanon, Israel, Cyprus, Türkiye, Syria, Iran and Afghanistan in Asia, northern Iraq and Pakistan, southeast Turkmenistan and Uzbekistan, and parts of Tajikistan and Kyrgyzstan. Egypt, Libya, Tunisia, and northern Algeria in Africa along the Mediterranean region, northern Morocco along the Mediterranean region, and western Morocco along the Atlantic region. Most of New Zealand and Tasmania in Oceania, as well as parts of southeastern Australia (Fig 8). The current suitable areas of *I. pacificus* is mainly concentrated in temperate marine climate, Mediterranean climate, and temperate continental climate along coastal areas.

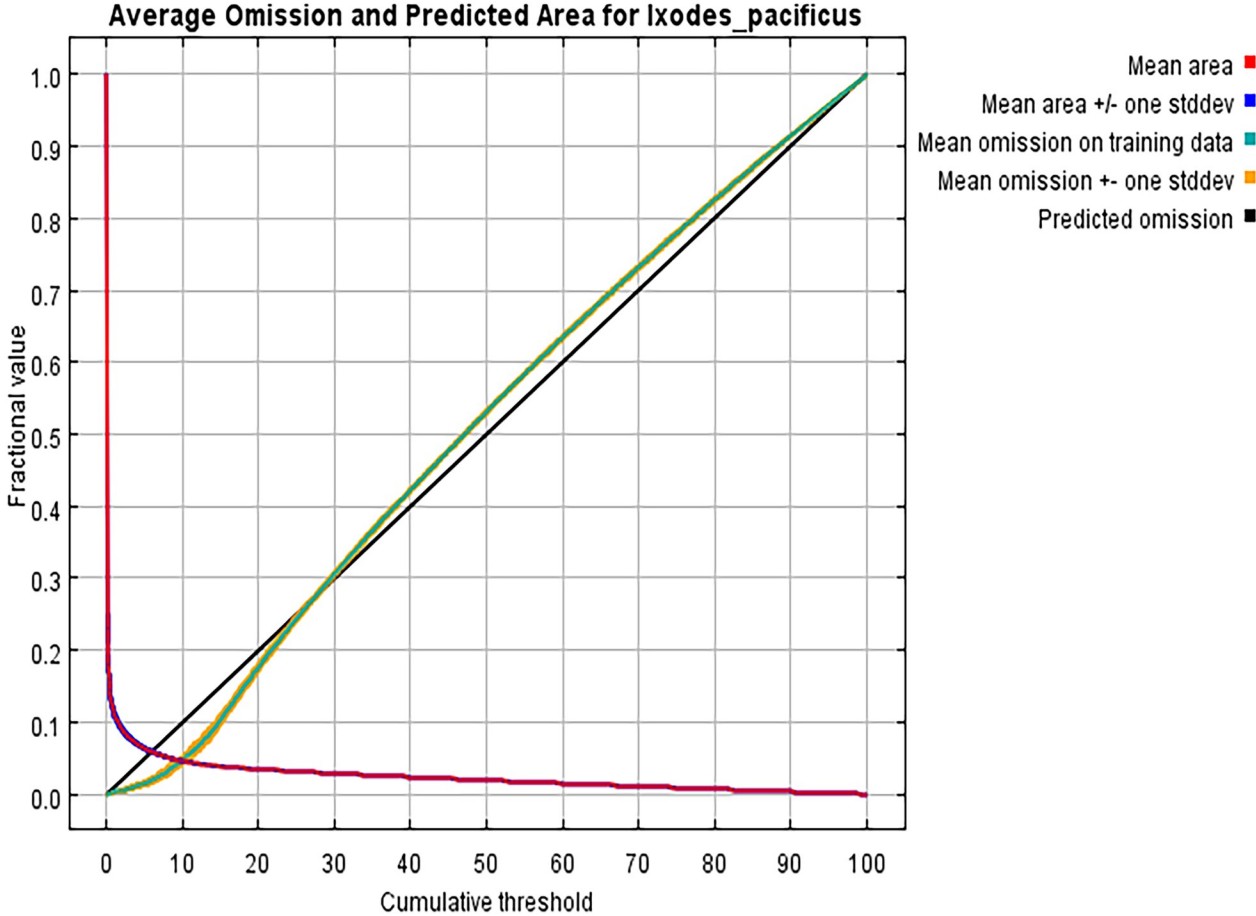

**Fig 4. Average omission and predicted area for *I. pacificus*.**

## The range of suitable areas for *I. pacificus* under future climatic conditions

According to the prediction results of the Maxent model, and visualize it using ArcGIS 10.8. Under future climatic conditions from 2021 to 2100. The suitable areas for *I. pacificus* in the world are similar to the near current distribution, but there are also certain differences, and the overall trend tends to shrink (Figs 9–12).

Under the climatic conditions of shared socio-economic pathway 1–2.6, the suitable areas of *I. pacificus* showed an expansion phenomenon from 2041 to 2060 compared to the near current climatic conditions, with an area change rate of 1%. In other time periods of the SSP1-2.6 scenario, as well as in the SSP2-4.5, SSP3-7.0, and SSP5-8.5 scenarios from 2021 to 2100, the potential global suitable areas of the *I. pacificus* is smaller than that of the near current climate scenario, with an area change rate of -25.70% -1.66%. In the SSP1-2.6 scenario mode between 2041 and 2060, the potential global suitable areas of *I. pacificus* reached a maximum area of $12.17 \times 10^6$ km$^2$. In the SSP3-7.0 and SSP5-8.5 scenarios, as time goes on, the global habitat range of the Pacific tick gradually decreases (Table 3). In the SSP5-8.5 scenario mode from 2081 to 2100, the potential global suitable areas of the *I. pacificus* decreases by $3.10 \times 10^6$ km$^2$. The global potential habitat area of the *I. pacificus* is the smallest, about $8.95 \times 10^6$ km$^2$.

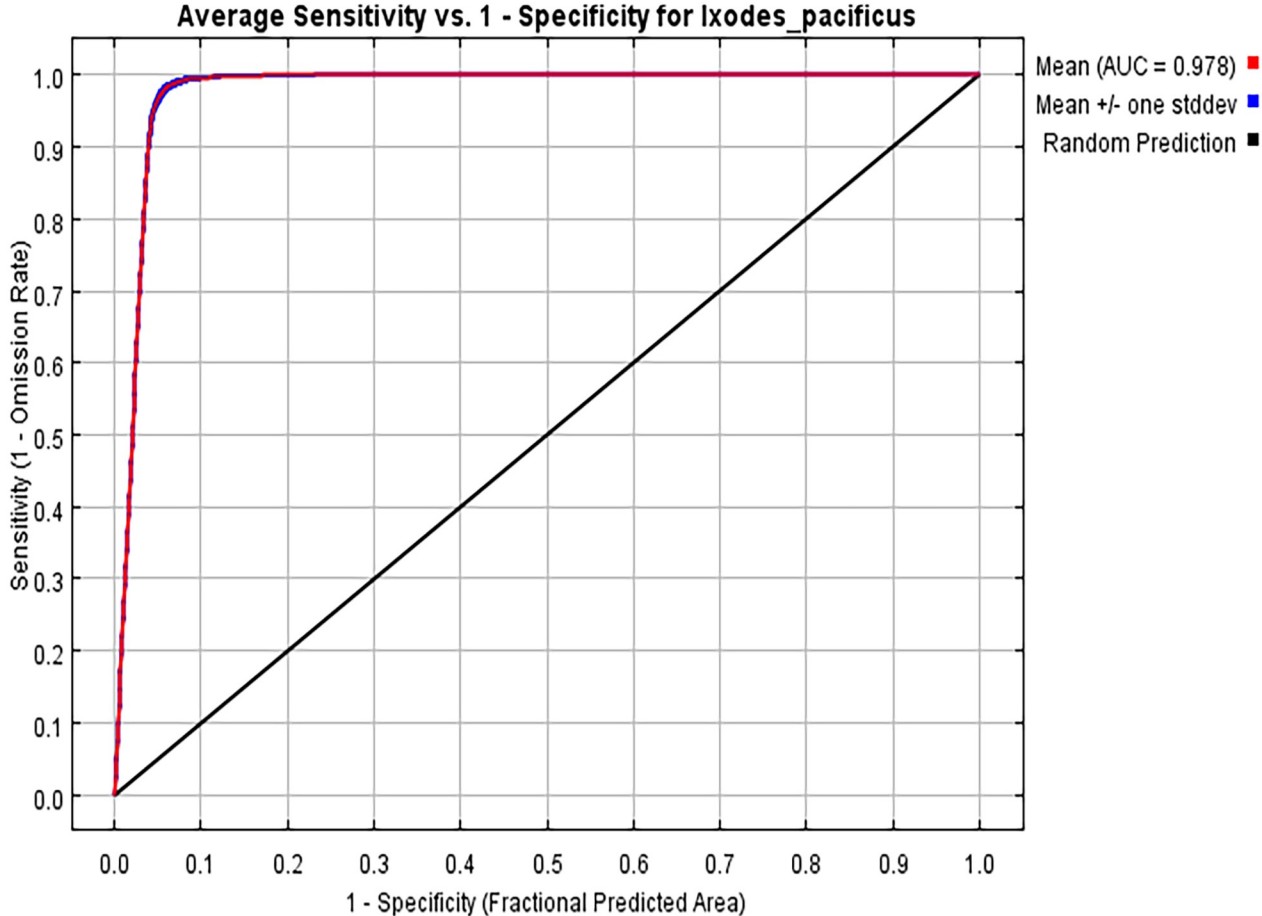

**Fig 5. Receiver operating characteristic curve of model prediction results.**

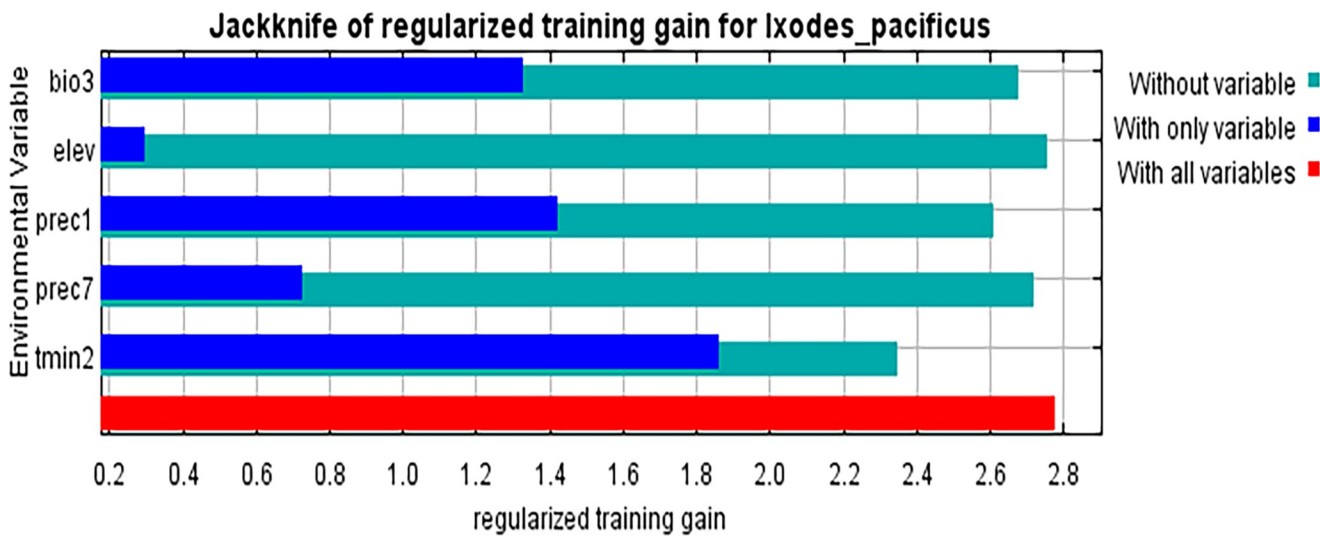

**Fig 6. Importance of the influence of environmental variables on the distribution of *I. pacificus*.**

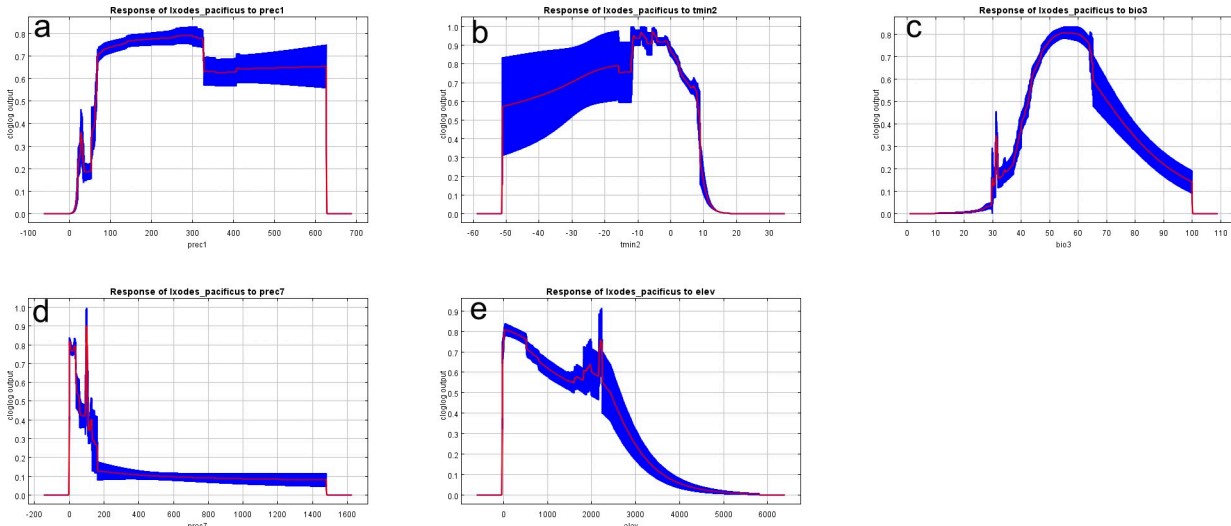

**Fig 7. Response curves of climatic variables to the distribution probability of *I. pacificus*.**

## Discussion

The distribution of *I. pacificus* is related to climate variables. Ticks generally go through four life stages: egg, larva, nymph, and adult. Most of their life cycle is exposed to the natural environment, so ticks need to adapt to local non biological factors (humidity, soil moisture and temperature.) and biological factors (damp leaves, dense vegetation, dense tree shade and host interactions.) [16]. When biting hosts to feed on blood, ticks are in a low metabolic state. At this time, ticks are particularly sensitive to environmental conditions such as temperature and humidity, and use vegetation and fallen leaves to regulate their microenvironment [17]. In

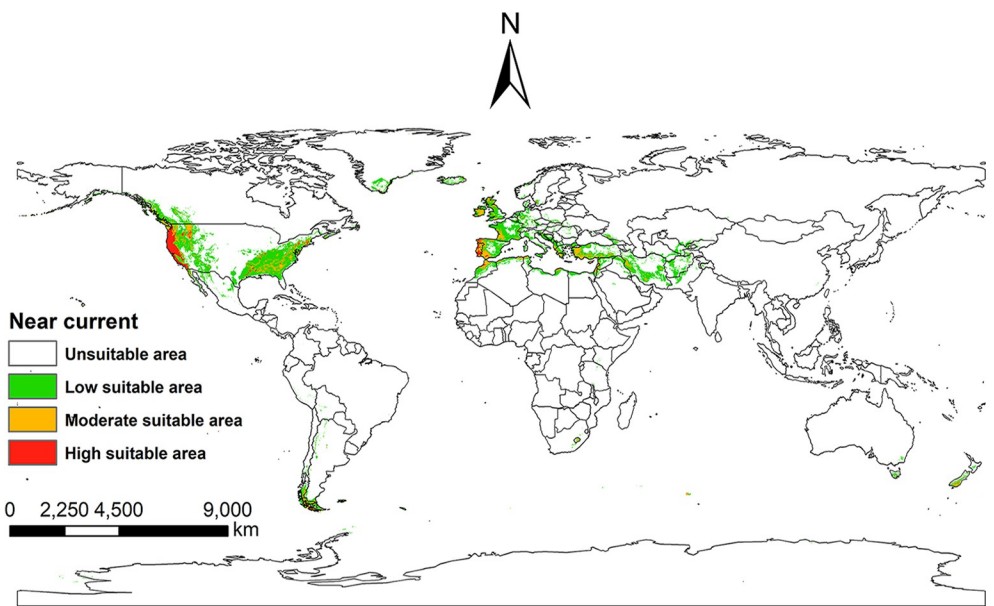

**Fig 8. Global potential suitable areas of *I. pacificus* under near current climatic conditions.** Made with Natural Earth. Free vector and raster map data @ naturalearthdata.com.

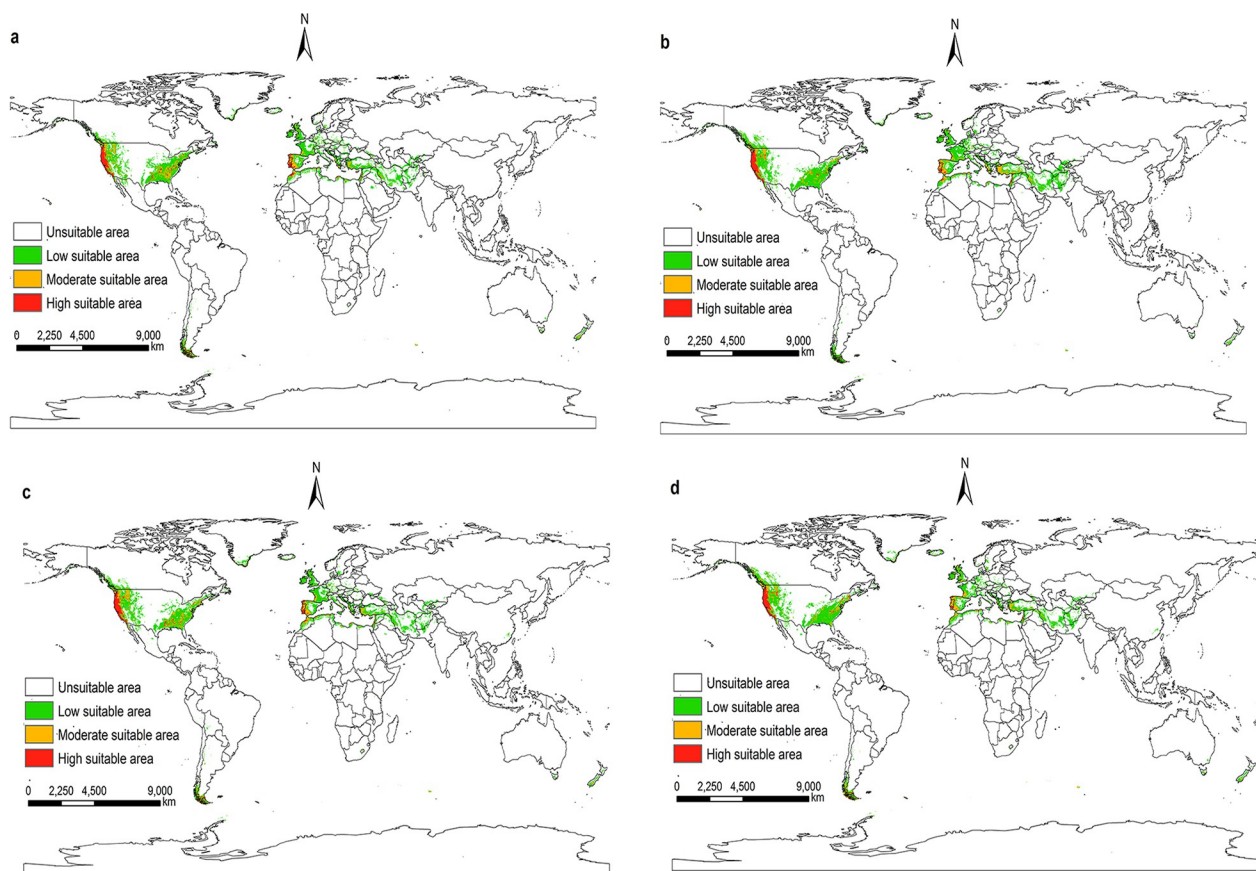

**Fig 9. The potential distribution of suitable areas for *I. pacificus* around the world under the climatic conditions of SSP1-2.6.** (a)2021-2040; (b) 2041-2060; (c)2061-2080; (d)2081-2100. Made with Natural Earth. Free vector and raster map data @ naturalearthdata.com.

addition to directly affecting the survival of ticks, climate conditions also affect the growth rate of ticks in immature life stages and seasonal behavior of searching for hosts [18]. Related studies on modeling non biological and biological factors related to the distribution of *I. pacificus* have shown that temperature and precipitation in the cold season are important factors affecting the environmental suitability of *I. pacificus*, with warm and humid climate conditions in winter being the most suitable [3, 19]. The results are basically consistent with the results of this study. The results of this study show that the main climate variables affecting the distribution of *I. pacificus* are Prec1, Tmin2, Bio3, Prec7 and Elev, among which Tmin2 and Prec1 contribute the most. When the lowest temperature in the cold season is between -45.1–7.9˚C, it may be more suitable for the survival of *I. pacificus*. When the winter precipitation is between 70-626mm, it is more suitable for the survival of *I. pacificus*, and the most suitable precipitation is 300mm.

The results of this study showed that the distribution range of *I. pacificus* showed a decreasing trend. Eisen et al. (2016) found that the distribution of *I. pacificus* has been relatively stable over the past two decades, which may indicate that the actual ecological niche of the current region of the *I. pacificus* is close to the range of its basic ecological niche (all countries and regions suitable for the survival of *I. pacificus* based on local environmental and climatic conditions). The basic niche may be wider than it actually is, but biological factors such as host migration or lack of hosts slow the spread of the tick range [19, 20]. Porter et al. (2021) used the future climate prediction (2050, RCP8.5) and found that about 30% of the habitat range of

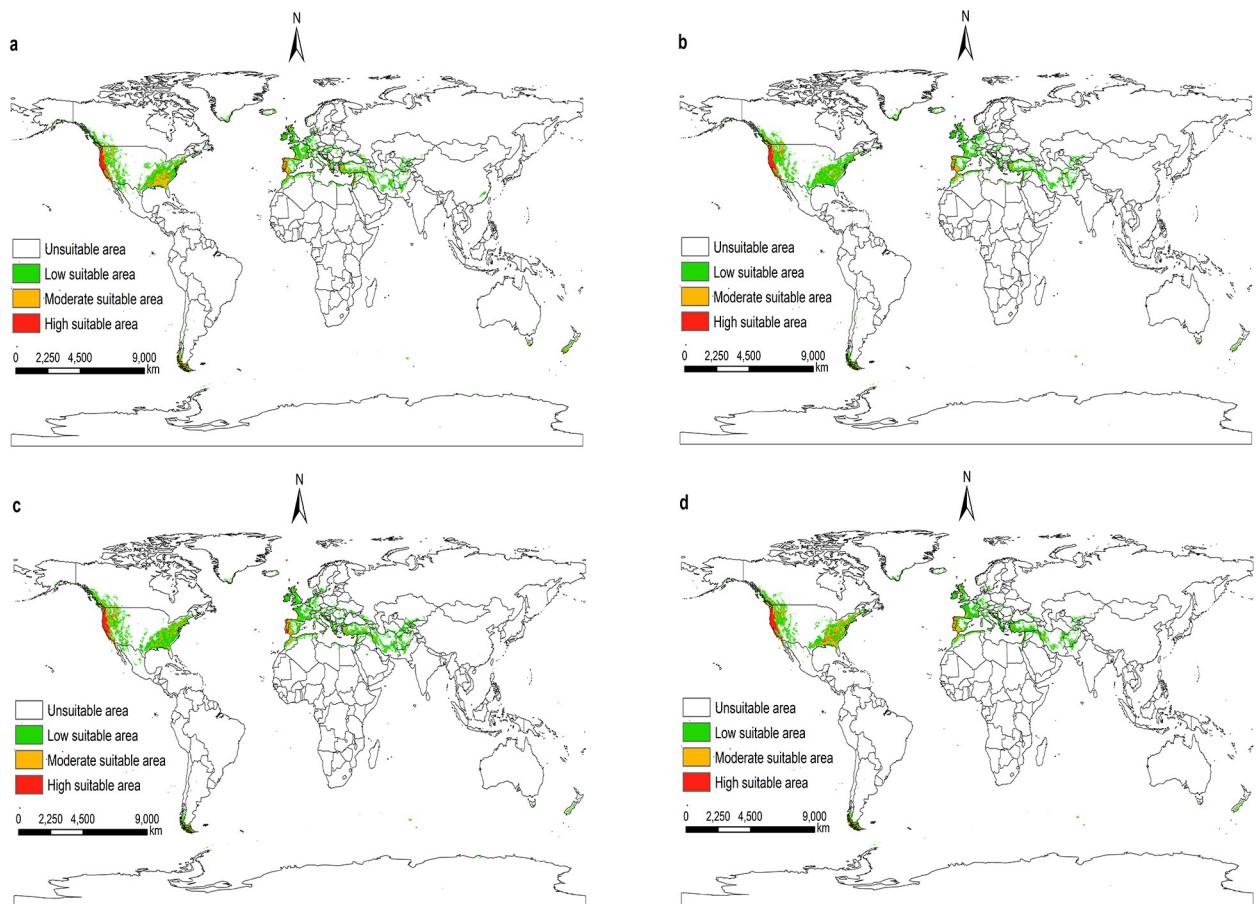

**Fig 10. The potential distribution of suitable areas for *I. pacificus* around the world under the climatic conditions of SSP2-4.5.** (a)2021-2040; (b)2041-2060; (c)2061-2080; (d)2081-2100. Made with Natural Earth. Free vector and raster map data @ naturalearthdata.com.

*I. pacificus* was lost in the western region of the United States [21]. MacDonald et al. (2018) found that the future habitat and suitable microclimate of *I. pacificus* will decrease overall, while the future habitat of *Dermacentor variabilis* and *Ixodes scapularis* will expand. The obvious differences in tick species distribution observed in this study indicate that global changes will not produce consistent results on vector-borne disease risk [22]. Alkishe et al. (2021), Hahn et al. (2021) and Witmer et al. (2022) showed that the distribution of *I. pacificus* is predicted to change under a warming climate, and the distribution of ticks in the western United States may expand to high elevations in the mountains of California, Oregon, and Washington. It extends north along the Pacific coast of British Columbia to the southernmost tip of Alaska, but in a warmer climate, the southernmost tip of California (including the mountains of the southern coast and the southern interior) and inland areas of Arizona and Utah will be inhospitable to *I. pacificus* [23–25]. These related studies are basically consistent with the results of this study. The results of this study show that *I. pacificus* will share different social and economic paths in the future. Compared with the suitable areas obtained by the current climate, under the SSP1-2.6 scenario model, the distribution range of *I. pacificus* will expand from 2041 to 2060, but the change is small (1%). Under other climate scenarios, the distribution range of *I. pacificus* showed a shrinking phenomenon. Under SSP5-8.5 scenario, the distribution range of *I. pacificus* decreased by 25.7% from 2081 to 2100. With the development of global economy and trade, human activities are becoming more and more frequent, which have an impact on

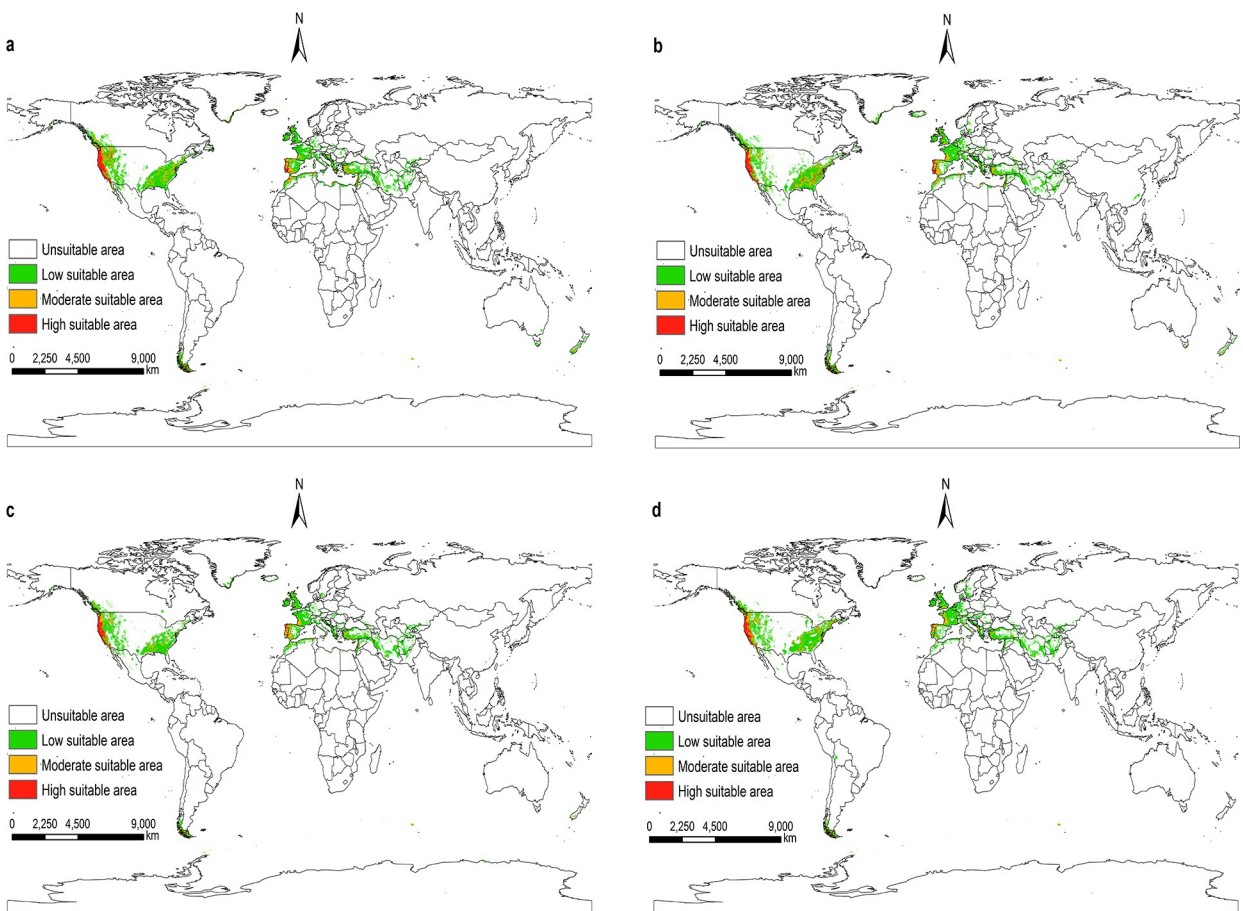

**Fig 11. The potential distribution of suitable areas for *I. pacificus* around the world under the climatic conditions of SSP3-7.0.** (a)2021-2040; (b)2041-2060; (c)2061-2080; (d)2081-2100. Made with Natural Earth. Free vector and raster map data @ naturalearthdata.com.

the natural environment, forest vegetation, mountains and deserts, the distribution of animals and plants, etc., and the global climate has also undergone a series of changes. In an environment where current environmental conditions are basically or almost suitable for the survival and reproduction of *I. pacificus*, future changes in precipitation and temperature patterns will have a significantly reduced effect on the distribution of *I. pacificus*.

Although the suitable areas of *I. pacificus* shows a trend of shrinking under the future climate scenario, the countries and regions involved in the potential medium and high suitable area of *I. pacificus* do not change much, mainly in the western and eastern parts of North America and the southwestern part of Canada. In addition, the results show that there are also suitable areas for *I. pacificus* in other countries and regions under the future climate change. For example, Chile and Argentina in South America; Most countries of Western Europe along the Atlantic and Mediterranean coasts; Most of New Zealand and Tasmania in Oceania, and parts of southeastern Australia. In Asia, Lebanon, Israel, Cyprus, Turkey, Syria, Iran, Afghanistan and other regions; In northern Africa along the Mediterranean and in southern Lesotho. And the preferred habitat of *I. pacificus* includes moist coastal redwood forest, dry oak or mandrone woodlands, mixed hardwoods, dense woodlands carpeted with leaf litter, grasslands, chaparral, and aspen forest [2]. Therefore, the invasion risk of *I. pacificus* should be focused on in coastal countries and regions with developed tourism or frequent international trade.

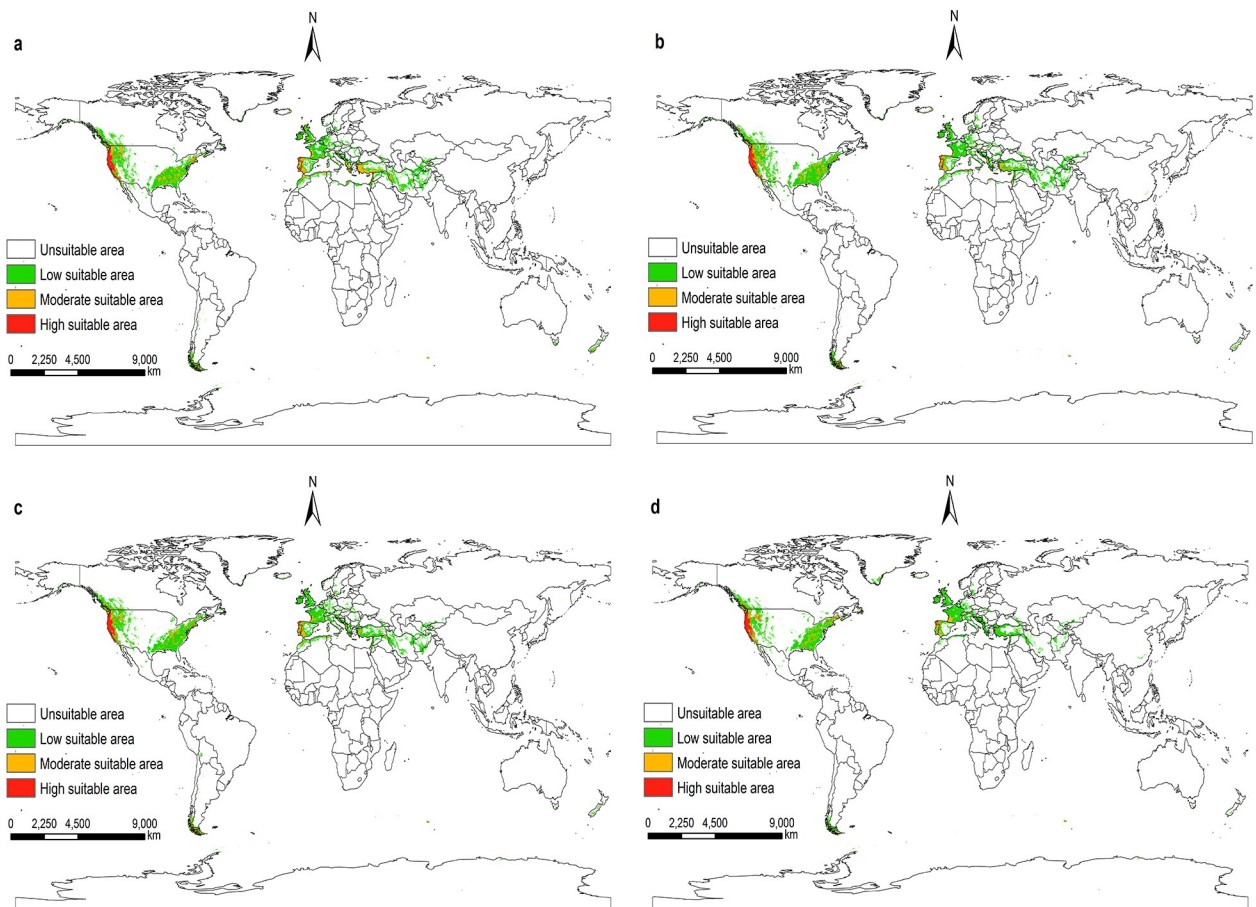

**Fig 12. The potential distribution of suitable areas for *I. pacificus* around the world under the climatic conditions of SSP5-8.5.** (a)2021-2040; (b)2041-2060; (c)2061-2080; (d)2081-2100. Made with Natural Earth. Free vector and raster map data @ naturalearthdata.com.

The survival ability of vectors such as *I. pacificus* depends on environmental conditions such as abiotic factors and biological factors. However, this study only considered the impact of climate on the distribution of *I. pacificus*, and did not introduce other abiotic factors such as vegetation type, land cover and host interaction. Some studies have shown that forest cover and land cover change are important factors for predicting the distribution of *I. pacificus* [3], which leads to certain limitations in this study and ignores the possible impacts of host migration and other factors on the distribution of *I. pacificus*. Global climate change may cause extreme weather such as droughts and floods, which has a great uncertainty on the distribution of *I. pacificus*.

The overall risk of tick-borne diseases depends on the ecological environment of the tick vector population and varies in response to global change processes depending on the region and vector species [22, 26, 27]. Under the condition of global climate change, climate warming and other changes may change the distribution pattern of disease vectors such as ticks, and the identification of environmental impact factors affecting the distribution of ticks has become an urgent public health issue. Vector surveillance, combined with habitat modeling, can provide a useful public health tool for detecting new areas of tick invasion and potential human risks [28–30]. This can be used to predict the possible distribution areas of vectors under future climate change and when large-scale field sampling is limited, to better provide effective

**Table 3. Current and future suitable area for *I. pacificus* across the world under different climatic conditions (×10⁶ km²).**

| Climate Scenario | Period | Less Suitable Areas | Moderately Suitable Areas | Highly Suitable Areas | Total Area | Area Change | Area Chang Ratio (%) |
|---|---|---|---|---|---|---|---|
| current | 1970–2000 | 8.93 | 2.43 | 0.69 | 12.05 | 0.00 | |
| SSP1-2.6 | 2021–2040 | 8.49 | 2.20 | 0.33 | 11.02 | -1.03 | -8.54 |
| | 2041–2060 | 9.45 | 2.06 | 0.66 | 12.17 | 0.12 | 1.00 |
| | 2061–2080 | 8.49 | 2.23 | 0.73 | 11.45 | -0.60 | -4.98 |
| | 2081–2100 | 8.33 | 2.06 | 0.67 | 11.06 | -0.99 | -8.21 |
| SSP2-4.5 | 2021–2040 | 8.34 | 2.67 | 0.59 | 11.60 | -0.45 | -3.73 |
| | 2041–2060 | 8.36 | 1.71 | 0.56 | 10.63 | -1.42 | -11.77 |
| | 2061–2080 | 9.25 | 2.05 | 0.55 | 11.85 | -0.20 | -1.66 |
| | 2081–2100 | 7.40 | 2.17 | 0.66 | 10.23 | -1.82 | -15.09 |
| SSP3-7.0 | 2021–2040 | 8.52 | 2.22 | 0.67 | 11.41 | -0.64 | -5.31 |
| | 2041–2060 | 8.19 | 2.14 | 0.64 | 10.97 | -1.08 | -8.96 |
| | 2061–2080 | 7.64 | 1.46 | 0.44 | 9.54 | -1.43 | -11.86 |
| | 2081–2100 | 8.15 | 1.70 | 0.45 | 10.30 | -1.75 | -14.51 |
| SSP5-8.5 | 2021–2040 | 8.45 | 2.44 | 0.52 | 11.41 | -0.64 | -5.31 |
| | 2041–2060 | 8.99 | 1.91 | 0.50 | 11.40 | -0.65 | -5.39 |
| | 2061–2080 | 8.18 | 1.75 | 0.47 | 10.40 | -1.65 | -13.68 |
| | 2081–2100 | 7.09 | 1.38 | 0.48 | 8.95 | -3.10 | -25.70 |

information for the changing risks of vector-borne diseases in the future, and provide references for public health early warning and control of vector-borne diseases.

## Conclusions

We found that the Maxent model predicted potentially suitable areas of *I. pacificus* under near current and future climatic conditions based on the existing distribution of *I. pacificus*, which may enable the precise identification of important environmental variables driving the current and future potential geographic distribution of *I. pacificus*, and this was derived from temperature and precipitation as they are the important variables affecting the survival probability. Under future climate conditions, the total suitable area will decrease and the distribution area will shrink. But with global economic integration, *I. pacificus* will have more opportunities to be carried to many countries. This is of great guiding significance for improving the awareness of prevention in areas where *I. pacificus* is at risk of invasion, thus providing a theoretical basis for early prevention and control in these areas.

## Supporting information

**S1 File. The coordinates of all *I. pacificus* collected globally.**
(XLSX)

**S2 File. The corresponding value of the response curve of Prec1 to the distribution probability of *I. pacificus*.**
(XLSX)

**S3 File. The corresponding value of the response curve of Tmin2 to the distribution probability of *I. pacificus*.**
(XLSX)

**S4 File. The corresponding value of the response curve of Bio3 to the distribution probability of *I. pacificus*.**
(XLSX)

**S5 File. The corresponding value of the response curve of Prec7 to the distribution probability of *I. pacificus*.**
(XLSX)

**S6 File. The corresponding value of the response curve of Elev to the distribution probability of *I. pacificus*.**
(XLSX)

## Acknowledgments

The authors thank all participants in this study.

## Author Contributions

**Conceptualization:** Fengfeng Li, Qunhong Wu.

**Data curation:** Fengfeng Li, Qunzheng Mu, Delong Ma.

**Formal analysis:** Fengfeng Li.

**Funding acquisition:** Qunhong Wu.

**Investigation:** Fengfeng Li.

**Methodology:** Fengfeng Li.

**Project administration:** Fengfeng Li, Qunhong Wu.

**Supervision:** Fengfeng Li, Qunhong Wu.

**Visualization:** Fengfeng Li.

**Writing – original draft:** Fengfeng Li.

**Writing – review & editing:** Fengfeng Li, Qunhong Wu.

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
