## [Decision Letter · Decision Letter 0]

3 Jul 2024

PONE-D-24-17968Predicting the Potential Global Distribution of *Ixodes pacificus* under climate ChangePLOS ONE

Dear Dr. li,

Thank you for submitting your manuscript to PLOS ONE. After careful consideration, we feel that it has merit but does not fully meet PLOS ONE’s publication criteria as it currently stands. Therefore, we invite you to submit a revised version of the manuscript that addresses the points raised during the review process.

**ACADEMIC EDITOR: **The manuscript needs revisions specifically English language editing. 

We look forward to receiving your revised manuscript.

Kind regards,

Shawky M Aboelhadid, PhD

Academic Editor

PLOS ONE

Journal Requirements:

6. We note that [Figures 1, and 8-12] in your submission contain [map/satellite] images which may be copyrighted. All PLOS content is published under the Creative Commons Attribution License (CC BY 4.0), which means that the manuscript, images, and Supporting Information files will be freely available online, and any third party is permitted to access, download, copy, distribute, and use these materials in any way, even commercially, with proper attribution. For these reasons, we cannot publish previously copyrighted maps or satellite images created using proprietary data, such as Google software (Google Maps, Street View, and Earth). For more information, see our copyright guidelines: http://journals.plos.org/plosone/s/licenses-and-copyright.

a. You may seek permission from the original copyright holder of Figures 1, and 8-12 to publish the content specifically under the CC BY 4.0 license.  

Reviewers' comments:

Reviewer's Responses to Questions

**Comments to the Author**

1. Is the manuscript technically sound, and do the data support the conclusions?

Reviewer #1: Partly

2. Has the statistical analysis been performed appropriately and rigorously? 

Reviewer #1: Yes

3. Have the authors made all data underlying the findings in their manuscript fully available?

Reviewer #1: Yes

4. Is the manuscript presented in an intelligible fashion and written in standard English?

Reviewer #1: Yes

5. Review Comments to the Author

Reviewer #1: This manuscript overall gives a broad overview of potential future Ixodes pacificus distribution. To my knowledge, the statistics were performed properly and the data was represented correctly. The main issues I have with this publication is grammar/spelling/structure wise and with the discussion.

The discussion needs quite a bit of work in order to be considered publishable. While the modeling is showing a wide range of suitable temperatures and habitats, the authors need to be careful to not interpret the model in a literal sense. This is a theoretical view of I. pacificus distribution and does not suggest that this kind of widespread distribution will happen. For example, the temperature range seems to be extremely low and wide and biologically not suitable for I. pacificus survival (other than temperatures above 0 degrees C). Is there literature to suggest that I. pacificus can theoretically survive –45.1 C? I have not seen such research done. As you mention, ticks are very sensitive to temperature and humidity and in my personal experience and from what has been seen in the literature, they have a small window which is most suitable for them, and it does not go that low. With the extreme ranges you found, you need to address the fact that most of these values are probably not compatible with life. Additionally, it may be that some of these other countries and continents are suitable, but I. pacificus has not been found there (as far as I am aware) and has a very narrow range. Please make sure to say this is theoretical.

Relevant literature is cited. Make sure to format in text citations properly; just the name is not enough, you need the date of the publication. Format based on your citation style. Double check spelling and grammar. Italicize all scientific names. Make sure all fonts are the same size. Be sure of your data, don’t write things like “about [this much]”, “basically” etc.

Specific comments:

Line 108: What are the bioclimatic variables? Can you be more specific (i.e. have a supplementary table with this information)?

Line 131-132: It sounds like you only kept variables with an absolute correlation coefficient of ≥ 0.8, but the following sentence says you retained those less than 0.8. Did you mean to say excluded?

Line 137: Why did you use A. americanum points? To train the software?

Line 293: List date for the Eisen paper, not just the name.

Line 302-303: Check spelling on I. scapularis and spell out Dermacentor variabilis, not American dog ticks.

Line 334-335: Why should we focus on it? Is there a trade route or human travel which indicates I. pacificus could become a problem in these areas? Explain.

6. PLOS authors have the option to publish the peer review history of their article (what does this mean?). If published, this will include your full peer review and any attached files.

Reviewer #1: No

---

## [Author Response · Author response to Decision Letter 0]

19 Jul 2024

Dear Editor,

Thank you very much for your advice on our manuscript (Manuscript ID: PONE-D-24-17968) entitled “Predicting the Potential Global Distribution of Ixodes pacificus under climate Change”. We also thank the academic editor and reviewers for their constructive comments and suggestions. We have revised the manuscript accordingly, and all amendments are indicated in the file labeled 'Revised Manuscript with Track Changes'. In addition, our point-by-point responses to the comments are listed below this letter.

We hope that our revised manuscript is now acceptable for publication in your journal and look forward to hearing from you soon.

With best wishes,

Yours sincerely,

Fengfeng Li, PhD

Shandong Second Medical University, China 

The response for the academic editor and reviewers:

First of all, we would like to express our sincere gratitude to the academic editor and reviewers for their constructive and positive comments.

Replies to Academic Editor

Major issues

Response: Thank you for your insightful suggestion. I have made strict changes to the names of my manuscripts and documents according to the format of the manuscripts required by your journals.

Response: In my Methods section, I mainly use the MaxEnt software for modeling species niches and distributions by applying a machine-learning technique. Maxent is now open source, and no permits were required. The information can be found at https://biodiversityinformatics.amnh.org/open_source/maxent/

3. Please note that PLOS ONE has specific guidelines on code sharing for submissions in which author-generated code underpins the findings in the manuscript. In these cases, we expect all author-generated code to be made available without restrictions upon publication of the work. 

Response: The code generated by my manuscript can be provided without restriction when the work is published. I have reviewed your guidelines and ensured that the code is shared in a way that follows best practices and promotes reproducibility and reuse.

4. We note that your Data Availability Statement is currently as follows: [All relevant data are within the manuscript and its Supporting Information files.] Please confirm at this time whether or not your submission contains all raw data required to replicate the results of your study. Authors must share the “minimal data set” for their submission.

Response: I confirm that my submission contains all the raw data required to replicate my research results.

Response: According to your suggestion, I have created a new ORCID iD and linked it to my Editorial Manager account.

6. We note that [Figures 1, and 8-12] in your submission contain [map/satellite] images which may be copyrighted. All PLOS content is published under the Creative Commons Attribution License (CC BY 4.0), which means that the manuscript, images, and Supporting Information files will be freely available online, and any third party is permitted to access, download, copy, distribute, and use these materials in any way, even commercially, with proper attribution. For these reasons, we cannot publish previously copyrighted maps or satellite images created using proprietary data, such as Google software (Google Maps, Street View, and Earth).

Response: I have revised the manuscript in response to your question. The details are as follows.

Lines 91-92: change “Google Earth (https://earth.google.com)” to “OpenStreeMap (https://osm.openmaptiles.org/)”. We have replaced Google Earth with OpenStreeMap to check the coordinate points again. OpenStreetMap is open data, The information can be found at https://osm.openmaptiles.org/copyright/en. 

Figures 1, and 8-12: These map data are derived from Natural Earth, and the figure caption have been updated with source information. All versions of Natural Earth raster + vector map data found on this website are in the public domain. The information can be found at https://www.naturalearthdata.com/about/terms-of-use/

Replies to Reviewer 1

Specific comments:

1. Line 108: What are the bioclimatic variables? Can you be more specific (i.e. have a supplementary table with this information)?

Response: I have made a table to describe the bioclimatic variables. Line 129: table1.

2. Line 131-132: It sounds like you only kept variables with an absolute correlation coefficient of ≥ 0.8, but the following sentence says you retained those less than 0.8. Did you mean to say excluded?

Response: Variables with an absolute correlation coefficient of less than 0.8 are retained and incorporated into the model. 

3. Line 137: Why did you use A. americanum points? To train the software? 

Response: change “A. americanum” to “I. pacificus”.

4. Line 293: List date for the Eisen paper, not just the name.

Response: change “Eisen” to “Eisen et al. (2016)”.

5. Line 302-303: Check spelling on I. scapularis and spell out Dermacentor variabilis, not American dog ticks.

Response: change “Ixodes scapulare” to “I. scapularis”. change “American dog ticks” to “Dermacentor variabilis”.

6. Line 334-335: Why should we focus on it? Is there a trade route or human travel which indicates I. pacificus could become a problem in these areas? Explain.

Response: The results of my study show that in the context of future climate change, in addition to the western and eastern parts of North America and the southwestern part of Canada, other countries around the world also have suitable areas for I. pacificus, and the preferred habitat of I. pacificus includes moist coastal redwood forest, dry oak or mandrone woodlands, mixed hardwoods, dense woodlands carpeted with leaf litter, grasslands, chaparral, and aspen forest [1]. Therefore, the invasion risk of I. pacificus should be focused on in coastal countries and regions with developed tourism or frequent international trade. (Supplemented in the manuscript Line 349-351, Line 355-359)

References:

1. Davis RS, Ramirez RA, Anderson JL, Bernhardt SA. Distribution and Habitat of Ixodes pacificus (Acari: Ixodidae) and Prevalence of Borrelia burgdorferi in Utah. J Med Entomol. 2015 Nov;52(6):1361-7. doi: 10.1093/jme/tjv124. Epub 2015 Aug 17. PMID: 26336263

---

## [Editor Report · Decision Letter 1]

12 Aug 2024

Predicting the Potential Global Distribution of *Ixodes pacificus* under climate Change

PONE-D-24-17968R1

Dear Dr. Fengfeng Li,

We’re pleased to inform you that your manuscript has been judged scientifically suitable for publication and will be formally accepted for publication once it meets all outstanding technical requirements.

Kind regards,

Shawky M Aboelhadid, PhD

Academic Editor

PLOS ONE
---

## [Editor Report · Acceptance letter]

16 Aug 2024

PONE-D-24-17968R1 

PLOS ONE

Dear Dr. li, 

I'm pleased to inform you that your manuscript has been deemed suitable for publication in PLOS ONE. Congratulations! Your manuscript is now being handed over to our production team.

Kind regards, 

on behalf of

Professor Shawky M Aboelhadid 

Academic Editor

PLOS ONE